# (Epi)genetic Modifications in Myogenic Stem Cells: From Novel Insights to Therapeutic Perspectives

**DOI:** 10.3390/cells8050429

**Published:** 2019-05-09

**Authors:** Natacha Breuls, Giorgia Giacomazzi, Maurilio Sampaolesi

**Affiliations:** 1Translational Cardiomyology Lab, Department of Development and Regeneration, Stem Cell Institute Leuven, 3000 KU Leuven, Belgium; natacha.breuls@kuleuven.be (N.B.); Giorgia.Giacomazzi@ieo.it (G.G.); 2Human Anatomy Unit, Department of Public Health, Experimental and Forensic Medicine, and Interuniversity Institute of Myology, University of Pavia, 27100 Pavia, Italy

**Keywords:** epigenetics, skeletal muscle, genome-engeneering, stem cell therapy

## Abstract

The skeletal muscle is considered to be an ideal target for stem cell therapy as it has an inherent regenerative capacity. Upon injury, the satellite cells, muscle stem cells that reside under the basal lamina of the myofibres, start to differentiate in order to reconstitute the myofibres while maintaining the initial stem cell pool. In recent years, it has become more and more evident that epigenetic mechanisms such as histon modifications, DNA methylations and microRNA modulations play a pivatol role in this differentiation process. By understanding the mechanisms behind myogenesis, researchers are able to use this knowledge to enhance the differentiation and engraftment potential of different muscle stem cells. Besides manipulation on an epigenetic level, recent advances in the field of genome-engineering allow site-specific modifications in the genome of these stem cells. Combining epigenetic control of the stem cell fate with the ability to site-specifically correct mutations or add genes for further cell control, can increase the use of stem cells as treatment of muscular dystrophies drastically. In this review, we will discuss the advances that have been made in genome-engineering and the epigenetic regulation of muscle stem cells and how this knowledge can help to get stem cell therapy to its full potential.

## 1. Introduction

The skeletal muscle accounts for ~40% of the human body weight and is important for movement and stability. Each muscle is composed of multinucleated myofibres, which are formed through the fusion of myoblasts during development. During embryogenesis, the skeletal muscle originates from paraxial mesoderm, part of the mesoderm located at the posterior part of the embryo. At the posterior tip of the embryo, the paraxial mesoderm starts to condense into somites progressively from head to tail. The newly formed somites will then adopt a sclerotomal (ventral) or dermomyotomal (dorsal) fate. Afterwards, the central part of the dermomyotome, defined by paired box gene 3 (Pax3) expression, disintegrates and the muscle progenitors start to give rise to the myotome [1,2]. When myogenesis begins, Pax3 is downregulated and specific basic helix-loop-helix transcription factors, known as muscle regulatory factors (MRFs), are sequentially upregulated. Myogenic factor 5 (Myf5) and subsequently muscle determining factor (MyoD), known as muscle determination genes, are upregulated after migration into the myotome, resulting in the myogenic determination of the precursor cells. Afterwards, myoblast differentiation is mainly led by Myogenin [3] (Figure 1A).

During adult life, in vivo myogenesis is mainly regulated by satellite cells (SCs), myogenic progenitors that reside under the basal lamina of the muscle fibres in a mitotically quiescent state. These SCs express Pax7, an analogue of Pax3, important for the persistence of the SC pool during post-natal life. Upon injury, the SCs become active and start to differentiate through the sequential expression of Myf5, MyoD and Myogenin (Figure 1B). This ability to incorporate mononucleated myoblasts into the multinucleated fibres makes the skeletal muscle an ideal target for stem cell therapy [4,5,6]. In fact, when a genetically corrected stem cell fuses with the existing myotube, this feature will be transferred within the long multinucleated syncytial tissue.

Muscular dystrophies (MDs) are a group of neuromuscular disorders, primarily characterized by progressive muscle weakening. Most of these disorders are caused by a single mutation within the dystrophin-associated glycoprotein complex. This complex normally serves as a mechanical link between the cytoskeleton and the laminin of the extracellular matrix, thereby stabilising the fibres upon muscle contraction. A mutation leading to the absence or dysfunction of one of these proteins, renders the sarcolemma vulnerable to contraction-induced damage, further leading to progressive fibre damage [7]. The most common type is Duchenne MD (DMD), a childhood MD caused by a mutation in the *dystrophin* gene. So far, therapy is focussed on the amelioration of symptoms rather than treatment of the disease [8,9]. Due to the chronic nature of MD, the endogenous stem cell pool becomes exhausted (Figure 1B). Therefore, patients could benefit from stem cell therapy to replenish the stem cell pool and regenerate the muscle.

The first stem cell pools considered to reconstitute the skeletal muscle are myoblasts or SCs, responsible for muscle regeneration in a physiological setting. Although they have the inherent capacity to reconstitute the muscle fibres, these stem cells lose their engraftment potential rapidly when they are cultured ex vivo. Furthermore, these cells lack migratory capacity causing the need for high-density cell injections at multiple sites to reach significant engraftment [10,11]. Another inherent stem cell is the mesoangioblast (MAB), a vessel-associated stem cell that can differentiate into all tissues of mesodermal origin [12]. MABs have the benefit that they can migrate into the muscle once they are injected into the bloodstream. However, despite encouraging preclinical results, clinical trials failed to show any functional improvement so far, suggesting that there is room to enhance the myogenic potential of MABs [13,14,15].

A pool of stem cells that have gained interest in recent years are induced pluripotent stem cells (iPSCs). These cells have the benefit that they possess an unlimited in vitro proliferative capacity and have the ability to differentiate into all cell types. Nevertheless, researchers are still struggling to obtain a pure myogenic population from iPSCs in order to avoid the risk of uncontrolled cell growth, once injected in vivo.

## 2. Epigenetics

In order to improve the potential of the aforementioned stem cells, strategies can be developed through the recent insights in the regulation of endogenous myogenesis. A recent topic that has received much attention is the epigenetic regulation of skeletal muscle regeneration. Epigenetics comprises all heritable mechanisms that do not affect the DNA sequence itself. These epigenetic marks can lie on the DNA itself (methylation) or on the histones surrounding the DNA (methylation, phosphorylation, acetylation and ubiquitination of their amino acid residues). These marks will influence the configuration of the chromatin. When the DNA is loosely wrapped around the histones, due to activation marks such as acetylations, genes can be transcribed. When repressive marks, such as DNA methylations, are present, gene transcription in these areas is blocked [16,17]. Another mechanism by which gene expression can be repressed is through the post-transcriptional binding of microRNAs (miRNAs) to the mRNA. All these epigenetic regulations have been implemented in myogenesis and can be used to manipulate the potential of muscle stem cells.

### 2.1. Epigenetic Regulation of Myogenesis

#### 2.1.1. DNA Methylation 

DNA Methylation introduces a methyl group to the cytosine residue, thereby causing steric hindrance, which prevents DNA-binding proteins from binding. The two groups of enzymes responsible for these methylations are the DNA methyltransferases (DNMTs) and the ten-eleven translocation methylcytosine dioxygenase (TET) family of proteins [18,19]. CpG islands, areas of the genome rich in cytosine residues followed by guanine residues, are considered regulatory regions for DNA methylation. Although not all promoters have CpG islands, the hypermethylation of these regions is associated with gene silencing. A study that described the methylome changes during myobalst development, has reported the occurrence of hypermethylation waves during skeletal muscle-lineage commitment. Those regions remained stably methylated, whereas muscle-regulatory regions, including the master gene Myf5, underwent hypomethylation to allow the onset of the myogenic differentiation program [20]. Global DNA methylation is increased during the progression from myoblasts to mature myotubes. Furthermore, myogenic stem cell activation and differentiation is marked by the upregulation of specific DNMT isoforms [21].

#### 2.1.2. Histone Methylation

The chromatin is composed out of a chain of nucleosomes, which are, in turn, built up out of four histones- H2A, H2B, H3 and H4. The amino acid tails of these histones are subjected to various post-transcriptional changes, which can regulate gene expression or repression. Methylations of histons generally occours at the lysine and arginine residues at the N-terminals of H3 and H4. Methyl groups are transferred from S-adenosyl methionine to these residues by histone methyltransferases (HMTs) [22]. The marks most implicated during skeletal muscle development are the trimethylation of histone 3 lysine 4 (H3K4m3), associated with gene activation, and the trimethylation of histone 3 lysine 27 or lysine 9 (H3K27m3/H3K9m3), involved in gene repression.

In quiescent SCs, the Polycomb Repressive Complex 2 (PRC2) is expressed, resulting in the acquisition of H3K27m3 silencing marks throughout the genome [23,24,25]. In addition, the transcription start sites of Pax7, Myf5 and MyoD are marked by an H3K4m3. Although the methylation, as such, does not induce expression, it marks the genes for activation leading to a rapid transcriptional induction upon the right stimuli. Activation of the SCs is marked by an acquisition of the repressive H3K27m3, known to be dominant over the H3K4m3, at the level of Pax7. In contrast, H3K4m3 is enriched at the chromatin neighbouring MyoD and Myf5 [26,27,28,29,30]. Master regulator MyoD has been associated with several HMTs, thereby regulating the expression of downstream MRFs (Figure 1). For example, in activated SCs HMT G9a is recruited to MyoD by homeoprotein Msx1, leading to the enrichment of H3K9me2 marks [31,32]. Although G9a represses MyoD, its expression is not abolished completely (Figure 2A). When MyoD binds to the MRF promotors, co-binding with HTMs KMT1A and Suv39h1 prevents premature activation of myogenic genes by stimulating repressive histone marks (Figure 2B) [33,34,35]. Upon differentiation, MyoD can be recruited to the promoters of several MRFs through the increased accessibility caused by the SWItch/Sucrose NonFermentable (SWI/SNF) chromatin-remodelling complex [36,37]. Once differentiation is induced, MyoD together with HMTs Set7/9 and DN-JMJD2A, leads to the removal of repressive histone marks and the acquisition of H3K4m3 (Figure 2D) [38,39]. 

#### 2.1.3. Histone Acetylation

Acetylation of the lysine residues on the histones neutralizes the positive charge, thereby relaxing the chromatin. By relaxing the chromatin, transcriptional binding sites become more accessible, leading to increased gene expression. In general, two types of enzymes control acetylation: the histone acetyltransferases (HATs) and the histone deacetylases (HDACs). The HDACs can be subdivided into four classes of which class I is the most transcriptionally active. In dormant SCs, MyoD activity is prevented through the binding of repressor Snai1/2 to the promoters of multiple MRFs and the subsequent recruiting of HDAC 1 and 2, rendering the MRFs in a hypo-acetylated state (Figure 1B) [40,41]. Once the SCs become activated, the Snai1/2:HDAC1/2 complex is removed through binding of phosphor-retinoblastoma (pRb) [42]. In addition, HAT p300 acetylates both MyoD and the downstream MRFs when associated with pCAF (Figure 2A,D) [43,44].

#### 2.1.4. miRNAs

miRNAs are 20–22 nucleotides long non-protein coding RNA molecules primarily involved in post-transcriptional gene regulation. The biogenesis of miRNAs starts in the nucleus, where miRNAs are generally transcribed by RNA polymerase II in primary transcripts called pre-miRs and subsequently cleaved by the microprocessor complex into shorter precursor molecules called pri-miRs. Pre-miRs are then transported into the cytosol, where they are further cleaved into ~22 nt long double-stranded molecules by a complex that includes a RNase-III, Dicer. miRNAs are then ready to be loaded on the RNA-induced silencing complex (RISC); the guide strand is loaded while the star strand is generally degraded [45]. miRNAs-mediated gene regulation is generally achieved via complementary binding of the single-stranded miRNA molecules to specific mRNA sequences, thus targeting mRNA for degradation. miRNAs are considered as epigenetic regulators since they primarily modulate protein level without modifying the gene sequences [46]. Additionally, miRNAs directly target epigenetic regulatory mechanisms, such as DNA methylation and histone modifications. miRNAs can, in fact, interfere with the epigenetic machinery by targeting DNMTs as they do in some types of cancers [47,48], or by binding histone deacetylases HDAC1 and HDAC4 [49].

miRNAs are actively involved at various stages of embryonic and adult myogenesis. miRNAs are key players in tuning striated muscle formation, both at the early stages of development and by modulation of stem cells during adult life. A group of miRNAs, known as the myomiRs, have been characterized in depth and their role in orchestrating skeletal myogenesis is well established [50]. MiR-1 and miR-133 are the main members of the myomir family. Both these miRNAs intrinsically modulate proliferation and differentiation of skeletal muscle stem cell lineages, though the direct regulation of serum response factor (SRF) and myocyte enhancer factor 2(MEF2) [51]. Another well-known myogenic player is mir-206, a positive regulator of myogenesis by supporting SC differentiation and repressing many negative modulators of skeletal muscle differentiation [52]. Alongside the aforementioned miRNAs, several others have been described as dynamic players in myogenesis, such as miR-181, a broadly expressed miRNA, which contributes to myoblast differentiation by targeting a MyoD repressor in mammals [53], miR-24 and miR-27 [54,55]. Finally, miR-669a downregulation has been linked to the severe disease progression in *sarcoglycan beta-null* mice, a mouse model for limb girdle MD type 2E, and long-term overexpression of miR-669a improved significantly *sarcoglycan beta-null* muscle structure and functionality [56,57]. 

### 2.2. Epigenetics to Skew Skeletal Muscle Differentiation

Due to the increasing knowledge in epigenetic regulation of myogenesis, researchers started to consider the adaptation of epigenetic marks as a strategy to reprogram the stem cell fate. An initial strategy used is drugs that can specifically modulate the epigenome. These drugs have the benefit of already being in the clinic for many years for several cancers. A more novel strategy is the use of miRNAs to manipulate the gene expression.

#### 2.2.1. Epigenetic Drugs

An initial strategy to manipulate the stem cell fate is through the use of epigenetic drugs. The first epigenetic drug used to manipulate the epigenetic state in myogenesis or MDs are the HDAC inhibitors (HDACi) [58]. HDACi have been proven to functionally and histologically restore the muscle fibres in several mouse models of MDs [59,60]. These beneficial outcomes have led to the start of a clinical trial to evaluate the effect of HDACi Givinostat in DMD patients [61]. So far, in phase I and II clinical trials, Givinostat is proven safe and counteracts the histological progression of DMD [62]. Currently, a phase III clinical trial is ongoing. 

The effect on the muscle stem cells specifically, has mainly been investigated with Trichostatin A and valproic acid. More specifically, in vitro, these HDACi increase the fusion of C2C12 myoblasts into myotubes [63]. Mechanistically, this effect is most probably mediated through the activation of MyoD and SNF/SWI complex protein Baf60c, leading to full epigenetic reprogramming [37,64]. In the case of valproic acid, this epigenetic reprogramming was even strong enough to induce myocyte differentiation from PSCs without the need for prior mesoderm formation or to directly transdifferentiate fibroblast and adipose-derived stem cells [65,66,67]. These findings indicate that these molecules can boost the myogenic potential of several sorts of stem cells in order to increase their in vivo efficacy.However, it should be noted that in adult stem cells, the effect of HDACi is stage-specific. Iezzi et al. showed that when C2C12 myoblasts are exposed to an HDACi prior to differentiation, the amount of myotube formation increases. In contrast, C2C12s differentiated together with an HDACi, lead to a decreased myotube formation [68]. These findings explain why some studies found a detrimental effect of HDACi on the myogenic differentiation [69,70].

Instead of promoting acetylation, the same effect can be reached by demethylating the DNA. Interestingly, the modulation of methylation via prolonged treatment with 5-azacytidine, an inhibitor of DNA methylation, has been associated with an increased myogenic commitment of fibroblasts, mature adipocyte-derived dedifferentiated fat cells and cardiac cells [71,72,73,74]. Furthermore, in C2C12s, the treatment with 5-azacytidine resulted in enhanced expression of muscle-specific genes (including myogenin), increased myotube maturation and spontaneous contraction, suggesting that inhibition of methylation could be a suitable tool to further boost myogenic differentiation in already committed precursors [75,76]. However, in more immature stem cells, such as embryonic stem cells or mesenchymal stem cells, DNA demethylation seems to favour differentiation towards the cardiac lineage [77,78,79].

#### 2.2.2. miRNA Modulations

Given their driving role in post-transcriptional gene regulation, miRNAs can contribute to epigenetic signalling and participate in the control of cell fate decisions. Moreover, their application potential is appealing thanks to their intrinsic properties to modulate several targets at once. In this view, manipulation of miRNAs to boost stem cells differentiation has been investigated. The traditional myomiRs have all been implicated in the enhancement of myoblasts differentiation. Studies have reported that the overexpression of miR-1 and miR133 enhances myogenic differentiation of murine SCs [64]. At the same time, the pivotal role of miR-206 in skeletal muscle regenerative strategies has been confirmed by in vitro studies on C2C12s and in vivo studies on an animal model of skeletal muscle degeneration [80]. Furthermore, the implementation of miRNAs in stem cell differentiation has been reported for cardiac regeneration as well. Here myomiRs, together with a cardiac-specific miRNA, miR-208, have been used to directly reprogram cardiac fibroblasts to cells with cardiomyocytes-like properties, pointing at the synergistic effect of multiple miRNAs in driving stem cell fate [81]. 

More recent work has shown evidence for the use of miRNA mimics or antagomiRs (or anti-miRs, chemically engineered oligonucleotides that prevent miRNA binding) to enhance differentiation efficacy in iPSCs by directly targeting gene expression. Inhibition of miR121 and miR122, using antagomiRs, has resulted in increased differentiation of human iPSCs to hematopoietic lineages by acting on the stem cell factor (SCF)/c-KIT signalling pathway [82]. In other studies, researchers have shown that miRNAs can be employed to modulate maturation of human iPSCs derived cardiomyocytes. A study conducted on iPSC differentiation has reported that members of the miR-290 cluster (miR-291-3p, miR-294, and miR-295) are directly involved in murine iPSC reprogramming, and additionally can influence cardiomyogenic lineages maturation [83]. Interestingly, this study pointed to an additional role of miRNAs, not only in downstream lineage differentiation but also upstream, at the reprogramming stage. miRNAs are, in fact, intrinsically programmed to play a role in crucial aspects of reprogramming, as for example, chromatic remodelling, achieved through the targeting of HDAC and DNA demethylation [84]. Finally, another study has reported the use of miRNAs derived from mesenchymal stem cells to help the maturation of iPSCs derived cardiomyocytes [85]. 

With regards to skeletal muscle differentiation, miRNAs can be involved in the specification of fibre types when iPSCs are differentiated to myogenic lineages [86]. The synergistic effect of miRNAs has been employed to drive the myogenic commitment of human iPSC-derived progenitors, strengthening the potential application of miRNAs in guiding lineage fate decisions [87]. In recent years, a study has again reported a role for miRNAs in refining reprogramming of somatic cells to iPSCs: selectively blocking age-induced miR-195 resulted in efficient iPSC generation from elderly donor subject [88]. These last two studies highlight the role of miRNAs in rescuing a gap when generating iPSCs progenitors that is dependent on the somatic cell source, further positioning miRNAs in the whereabouts of stem cell fate decision processes.

## 3. Genetics

Not only on an epigenetic level but also genetically, advances have been made that could aid the future of stem cell therapy. Engineering of the human genome consists of insertion, deletion, modification and/or replacement of pieces of DNA. Advances in the genome-editing technologies allow us to gain precise control over the target site [89].

The first approach to manipulate the genome was through random insertions by viral vectors or plasmids [90]. However, this random integration can lead to disruption of the physiological gene expression—a phenomenon called insertional mutagenesis. When the mutagenesis disrupts a tumour suppressor gene or activates an oncogene, it can lead to the development of cancer [91,92]. The most well-known example is the clinical trial where gene therapy in patients with X-linked severe combined immunodeficiency led to the development of leukaemia in multiple cases [93].

To overcome this risk, later genome-engineering techniques use programmable endonucleases to induce double-stranded breaks in a targeted manner. These breaks will activate DNA repair mechanisms such as non-homologous end-joining (NHEJ) or, when a template is provided, homology-directed repair (HDR) [89]. The most frequently used endonucleases are zinc-finger nucleases (ZFNs), transcription activator-like effector nucleases (TALENs) and the Clustered Regularly Interspaced Short Palindromic Repeats and CRISPR associated proteins (CRISPR/Cas) system [94,95]. Although their targeting efficiency is quite similar, nowadays the CRISPR system is the most broadly used as it has reduced cytotoxicity, a broader target range and is the easiest to assemble compared to the other methods [95,96]. Initially, the CRISPR-system was identified in numerous archaea and bacteria as part of the adaptive immune system. The CRISPR consist of two components: a single guide RNA and a non-specific CRISPR-associated endonuclease (Cas). The Cas protein will cleave the DNA when the guide RNA binds to a specific region in the DNA that is flanked by a protospacer-adjacent motif (PAM) sequence. The most studied CRISPR is the type II CRISPR/Cas9, originating from the *Streptococcus Pyogenes*. With this system, the Cas9 endonucleases can be directed towards a specific genomic locus through a single guide RNA sequence followed by the canonical PAM sequence 5′-NGG-3′ or 5′-NAG-3′ [97,98].

As MDs are mostly caused by a single gene mutation, genome engineering can be applied with the goal to restore certain genes that are affected. This would make it possible to derive stem cells from the patients themselves, thereby avoiding an immune response. Furthermore, genes that can help to control the cell fate or allow stem cell monitoring, can be inserted in a targeted manner. Here, we will discuss the most used option in which genome-engineering can be applied.

### 3.1. Correction of the Disease-Causing Mutation

Risk of immune rejection is a major hurdle that heterologous stem cell therapy is facing. This issue can be overcome by using the patients’ own cells. However, in order to use patient-specific cells, the underlying mutation needs to be corrected.

So far, most corrections have been performed on cells from patients with DMD (Table 1). DMD can be caused by deletions (~68%), duplications (~11%) or point mutations (~20%) in the *dystrophin* gene. Although these mutations can occur in the entire gene, some mutational “hotspots” are located between exons 2–19 and exons 45–55 [99]. When this mutation does not disrupt the reading frame, it results in a shorter but still functional protein, leading to a milder phenotype known as Becker MD. However, if the reading frame is disrupted or a premature stop codon is generated, the dystrophin will be completely absent, resulting in DMD [100] (Figure 3).

The dystrophin gene is the largest gene in the human genome, spanning more than 2.5 million base pairs [101] (Figure 3A). The large size of the gene renders it impossible to insert the complete gene into the genome. Alternative strategies include the use of truncated versions or dystrophin analogues. However, these additions can restore the function of dystrophin only partially [102,103,104]. Therefore, correction of the mutated gene can offer a solution.

In general, there are four different strategies that can be used to correct the DMD mutation, depending on the type of mutation and its location [105] (Table 1). The most traditional strategy to restore the reading frame is by an in-frame deletion of one or more exons [106,107,108]. Instead of completely deleting an exon, the splice donor or acceptor sequence can be removed, leading to permanent exon skipping [106,109,110,111,112]. However, unless an exon duplication has occurred, both these strategies will produce a truncated version of the protein, leading to Becker MD (Figure 3C). In order to minimise the length of the genomic deletion, a double-stranded break can be created by NHEJ in order to restore the reading frame [111,113,114]. However, this strategy is only successful in some cases as only one-third of the deletions created by NHEJ are in-frame. Lastly, the DMD mutation can also be corrected by the knock-in of an exon which will lead to the expression of the full-length dystrophin [111,115].

It should be noted that the site-specific correction of the DMD mutation has not only been attempted in stem cells ex vivo but also directly in vivo [116,117]. This strategy has led to the successful restoration of dystrophin in both skeletal muscle and the heart of *mdx* mice, a mouse model of DMD, and the canine model for DMD [109,110,118,119,120]. Although successful, there are still issues that remain to be tackled such as immunogenicity towards the delivery method and genotoxicity. Furthermore, this strategy requires the injection of the endonucleases directly into the patient, making it impossible to control for any off-target effects [121]. This can be overcome by correcting the mutation ex vivo in the patients’ stem cells, making it possible to check for any off-target effect before re-injecting them.

The earliest attempts to restore the dystrophin gene ex vivo focussed on myoblasts, using either mega-nucleases, ZFNs or TALENs [107,115,122,123]. Although all strategies successfully restored dystrophin expression, only the myoblasts corrected with ZFNs were tested in vivo for their ability to restore the skeletal muscle [107]. Afterwards, all additional attempts used the CRISPR-Cas9 system to correct the DMD mutation in myoblasts [106,114,122,123,125]. A direct comparison between the CRISPR/Cas9 and TALEN by Maggio et al. showed that the CRISPR/Cas9 offers the highest targeting flexibility and the least off-target effects [122,123]. 

Later attempts to correct the DMD mutation were focused mainly on PSCs and their derivatives due to their self-renewal abilities, which allow long-term in vitro manipulation [126]. The first correction of the dystrophin in patient-iPSCs was reported in 2015, using the TALENs to guide homologous repair [111]. In this study, the TALEN approach was compared directly to the CRISPR-Cas9 system. Although both systems were able to restore the dystrophin expression, the CRISPR-Cas9 system has been associated with less off-target sites [111]. A similar direct comparison of the two techniques showed a better gene-editing efficiency of the CRISPR-Cas9 system compared to TALENs [127]. Unsurprisingly, all published attempts in PSCs used the CRISPR-Cas9 system to restore the dystrophin gene [108,122,123].

In order to avoid the need for a new CRISPR/Cas9 complex for each patient, some groups came with a strategy to correct a diversity of DMD mutations with a single CRISPR. By deleting a large fragment within a mutational hotspot, dystrophin expression can be restored for around 60% of the DMD patients [112,113]. All patient-derived iPSCs showed a morphological and functional improvement after correction, when differentiated towards both skeletal muscle and cardiomyocytes [108,111,112,113]. Although the CRISPR-Cas9 system is the most effective technique so far, other techniques are being optimised. An example recently tested in DMD, is the CRISPR from *Prevotella and Francisella 1* (CRISPR-Cpf1), a CRISPR system that shows greater flexibility in target sites and higher activation of the DNA repair mechanisms, thereby enhancing the efficiency [124]. Lastly, although most studies focus on DMD, the same gene-editing strategies efficiently correct the disease-causing mutation in other MDs, such as Limb-girlde MD type 2D and myotonic dystrophy [128,129].

### 3.2. Gene Addition

Besides the integration of the healthy gene within a certain disease, much attention is paid to the insertion of genes that can help to gain more insight and control over the stem cells and their fate. In order to avoid insertional mutagenesis, the gene integration can be targeted towards so-called safe-harbour loci. Genomic safe-harbours are regions in the genome that ensure stable expression of the transgene without the risk of interrupting the endogenous gene expression [130]. The most well-known safe-harbour loci in the genome are the adeno-associated virus integration-site 1 (AAVS1) and the C-C motif chemokine receptor 5 (CCR5) [131,132].

#### 3.2.1. Transcription Factors

One of the most used examples is the integration of transcription factors to improve skeletal muscle differentiation in pluripotent stem cells [133]. Over the years, the integration of key regulators of skeletal myogenesis, such as Pax7 and MyoD, has led to a reproducible and highly efficient in vitro differentiation [134,135,136,137]. As most of these transcription factors only need to be active temporarily, the Piggy-Bac transposase system has gained much interest. This system links the transcription factors to a transposon making the construct removable upon administration of a transposase. Excision of this system leaves no trace, avoiding the risk for insertional mutagenesis [138]. 

#### 3.2.2. Reporter Genes

Reporter genes encode for proteins that have a high affinity for a certain probe. Once delivered in vivo, the cells can be visualised upon administration of the matching probe, thereby allowing longitudinal monitoring of the cell survival [139,140]. In the field of MD, reporter genes are widely used on a preclinical level. Green fluorescent protein for immunofluorescence and luciferase for bioluminescence imaging are the most widely used [141,142,143]. However, recently, attempts have been made to integrate reporter genes compatible with imaging modalities that can be applied clinically such as positron emission tomography (PET) and single-photon emission computed tomography (SPECT) [142,144]. In order to move this application towards the clinic, site-specific integration is essential. So far, only one report shows the integration of these reporter genes in a site-specific manner [144]. More specifically, Holvoet et al. integrated the human sodium iodide symporter into the AAVS1 locus of embryonic stem cells through the use of ZFNs. This integration led to a stable expression of the reporter over time without influencing the functionality and differentiation potential of the cells. By combining targeted genome-engineering with stem cell imaging, the transition of stem cell therapy to a clinical setting will be greatly accelerated.

#### 3.2.3. Suicide Genes

The integration of suicide genes is of great importance when using PSCs, as differentiations are often impure leading to the risk for teratoma formation [145]. Suicide genes are used as safety switches in order to intervene when PSC-derivatives undergo excessive proliferation. So far, this system has been mostly investigated on a PSC-level, making its use to specifically eliminate cells from the skeletal and cardiac muscle less explored. The first suicide gene used in the area of MD, is the inducible Caspase 9 system, an enzyme that induces apoptosis upon administration of a chemical inducer of dimerisation. This system was used eight weeks after injection of mesodermal progenitors into the skeletal muscle, showing complete clearance of the injected stem cells [146]. Within the cardiac muscle, the human somatostatin receptor 2 was incorporated, which made it possible to eliminate the engrafted cells upon the administration of ^177^Lu-DOTATATE, a β^-^-emitting radioisotope specifically binding the integrated receptor. The benefit of this gene is that, just by adding a different radionuclide, it can also serve as a reporter gene. This means that by integrating a single gene, the injected stem cells can be monitored and, upon observation of an abnormal growth, can be eliminated [147].

## 4. Conclusions and Future Directions

We examined the convergence of genetics and epigenetics that act in concert to advance our knowledge on myogenic stem cells and accelerate the development of novel therapies. We provided an overview of the genetic and epigenetic advances that can result in novel therapeutic treatments for muscle disorders. Adult stem cells, including SCs and MABs can transfer some epigenetic marks to their differentiated counterparts but are extremely rare in skeletal muscles. On the other hand, it is known that during reprogramming, iPS cells retain some epigenetic memory from somatic sources and these epigenetic markers affect cell-based therapies. At the moment, we are facing technical challenges, including the possibility to generate fully mature somatic cells from iPS cells, routine methods to carry out epigenomic analyses from a single cell and precise computational procedures to handle big data. Thus, a better knowledge of the interplay among genetic and epigenetic factors combined with single cell analysis in order to improve stem cell-based therapies is mandatory. In this view, more and more therapeutic approaches using miRNA mimics or antagomirs combined with gene-editing and iPS cell technology are expected in the near future.

## Figures and Tables

**Figure 1 cells-08-00429-f001:**
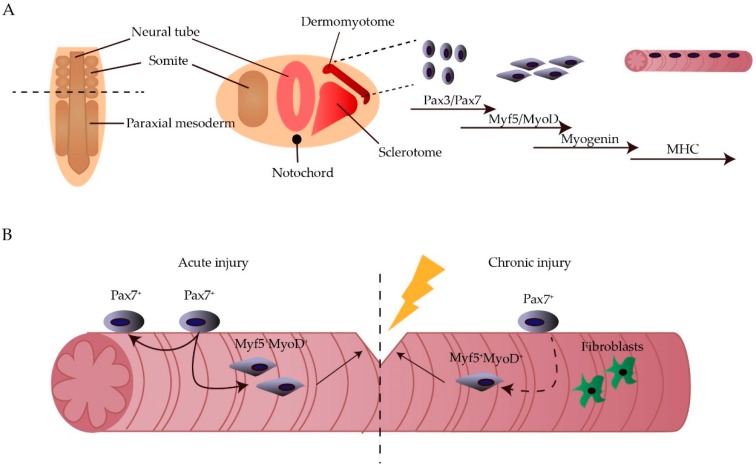
Myogenesis during embryonic development and adult life. During embryogenesis, the skeletal muscle originates from the paraxial mesoderm, which will further subdivide into somites. In the somites, Pax3^+^/Pax7^+^ cells will arise from the dermomyotome and will form muscle fibers through the sequential upregulation of Myf5, MyoD and myosin heavy chain (MHC) (**A**). During adult life, Pax7^+^ satellite cells are the main cell type responsible for muscle regeneration upon injury. However, when the injury is of chronic nature, this stem cell pool becomes exhausted and fibroblast are recruited to form scar tissue (**B**).

**Figure 2 cells-08-00429-f002:**
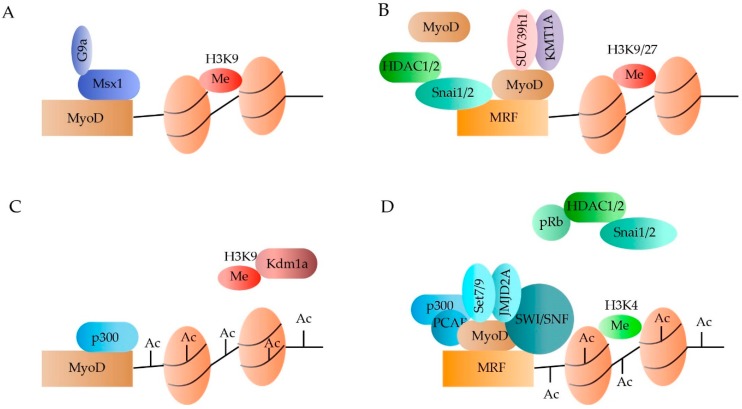
Epigenetic regulation of MyoD in the quiescent and activated satellite cells (SCs). MyoD is repressed in dormant SCs due to the presence of H3K9me2 marks, recruited by the Msx1/G9a complex (**A**). Furthermore, the binding of MyoD to the muscle regulatory factors (MRFs) through the binding of the Snai1/2:HDAC1/2 complex, renders the MRFs hypo-acetylated. The small amount of MyoD that binds to the MRFs will recruite SUV39h and KMT1A, leading to the addition of repressive H3K9/27m3 marks (**B**). Upon SC activation, repressive marks are removed from MyoD by Kdm1a and from the MRFs by phosphor-retinoblastoma (pRb). Furthermore, acetylation is increased by binding of p300 and pCAF (**C**,**D**). Once MyoD is activated, it will bind to the MRFs and recruit Set7/9, JMJD2A and SWI/SNF, which will add permissive H3K4m3 marks (**D**).

**Figure 3 cells-08-00429-f003:**
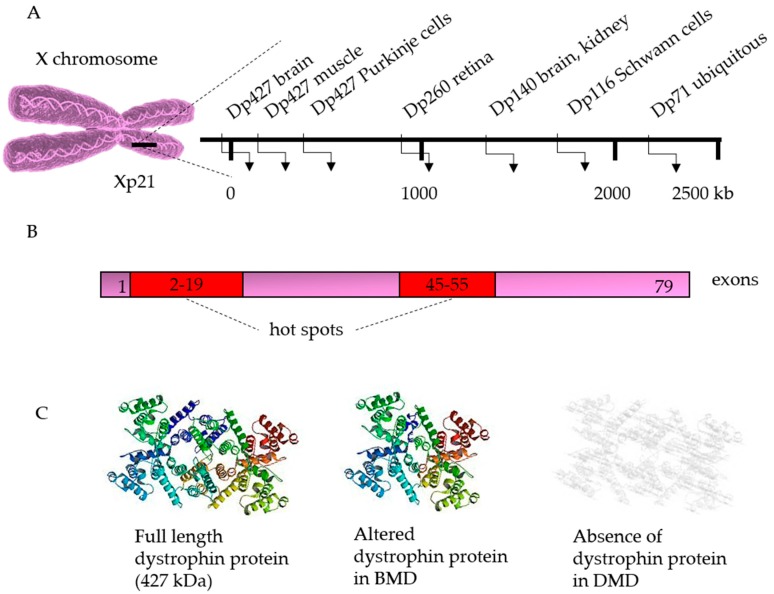
The dystrophin gene and its disease causing mutations. The dystrophin gene is located at chromosome Xp21 and presents several cell-specific promoters (**A**). The gene consists of 79 exons, distributed over about 2.5 million bases. In patients with Duchenne muscular dystrophy (DMD), protein translation is stopped prematurely due to the introduction of a stop codon or a frame-shift mutation. The major DMD deletion hot spots (in red) are located between exons 2–19 and between exons 45–55. Deletions are found in about 60–65% of patients, and the frequency of duplications may range from 5% to 15%. The remaining cases are caused by point mutations, intronic deletions, or exonic insertion of repetitive sequences (**B**). The full length dystrophin protein (427 kDA) contains an amino-terminal, a spectrin-like, a cysteine rich, and carboxy-terminal domains. In patients with Becker muscular dystrophy (BMD), mutations maintain the translational reading frame, generating a shorter but still functional dystrophin (**C**).

**Table 1 cells-08-00429-t001:** Ex vivo strategies for correction of mutations in the dystrophin gene.

Technique	Cell Type	Mutation	Correction Strategy	Delivery Method	Reference
Meganucleases	Myoblasts	ΔEx45-52	Knock-in Ex45-52	lentiviral	[115]
ZFN	Myoblasts	ΔEx48–50	Skip Ex51	Electroporation	[107]
TALEN	Myoblasts	ΔEx45-52	ΔEx44-54	Adenoviral	[122,123]
	iPSC	ΔEx44	Knock-in Ex44 Skip Ex44-45	Electroporation	[111]
CRISPR/Cas9	Myoblasts	ΔEx45-52	ΔEx53 ΔEx44-54 ΔEx51	Adenoviral	[122,123]
	Myoblasts	ΔEx45-52	Frameshift Ex51	Adenoviral	[122,123]
	Myoblasts	Dupl. Ex2	ΔDuplEx2	Lentiviral	[124]
	Myoblasts	ΔEx51-53	Reframing Ex50 and Ex54	Lipofectamine	[114]
	iPSC	ΔEx44	Knock-in Ex44 Skip Ex44-45 Frameshift Ex44	Electroporation	[111]
	iPSC	Δ48-50 Dupl. Ex55-59 Pt Ex47	Skip Ex47-52 ΔDuplEx55-59 Δmutated Ex47	Lipofectamine	[112]
	iPSCs	ΔEx8-9	ΔEx3–9, ΔEx6–9, or ΔEx7–11	Electroporation	[108]
	iPSC	Dup. Ex50 ΔEx46-51 ΔEx46-47	ΔEx45-55	Electroporation	[113]
CRISPR-Cpf1	iPSC	ΔEx48-59	Skip Ex51	Electroporation	[125]

ZFN: Zinc-Finger Nucleases; TALEN: Transcription Activator-Like Effector Nucleases; CRISPR: Clustered Regularly Interspaced Short Palindromic Repeats; Cas: CRISPR-associated Protein; Cpf: CRISPR from *Prevotella and Francisella 1*; Δ: Deletion; Ex: Exon; Dupl: Duplication; Pt: Point mutation.

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
