# Peer review of "(Epi)genetic Modifications in Myogenic Stem Cells: From Novel Insights to Therapeutic Perspectives"

_cells, 2019, doi:10.3390/cells8050429_

Reviewer 1 Report

The article by Breuls at al. provides a review of the epigenetic regulation of muscle stem cells and the potential use of epigenetic and genetic engineering approaches to develop stem cell therapy for muscular dystrophies. The subject of this review and the perspective given by the authors is of great interest for a broader lectureship of researches working in the field of cellular therapies, but also for a more general public. 

However, the paper has several important shortcomings, including generalisations, overstatements and even scientifically incorrect statements that should be corrected. In addition, the use of specialised language and lack of explanations for several abbreviations makes the understanding of several parts of the article difficult. Finally, the language needs extensive editing, as many mistakes are present. The review would also greatly benefit from more figures to visualise the concepts and details. In addition, the authors concentrate on the role of histone modifications and miRNA, but skip DNA methylation, the dynamic role of which in both myogenesis and muscular pathologies is well established. This should be either included (at least in a few sentences) or clearly stated that is out of scope.

Detailed points:

1) Several scientific statements are severely misleading or simply not true. The authors should  also avoid misleading generalisations.

l. 81 DNA can be methylated, but is not acetylated! Acetylation occurs on lysine residues of histones in the chromatin.

l.85 Histone methylation is not by definition repressing, the activating of repressing function of the methylation of the histone tails is context dependent (e.g. H3K4me3 is activating, while H3K9me3 is repressive). Similarly, miRNA can also activate gene expression (see for example PMID: 28639193). The authors should avoid misleading generalisations. 

l.94-95 'Methylation introduces a methyl group to the cytosine' in the context of histone methylation is WRONG. Cytosines are methylated in the DNA, in histones methylation occurs on lysine and arginines.

l. 97-98 DNMTs and TET enzymes are responsible for the methylation of DNA and NOT histones. Histones are methylated by histone (or protein) lysine/arginine methyltransferases and demethylated by histone (or protein) demethylases.

l. 123 'In general, the genome is in a deacetylated state'. This is again a generalisation. Do the authors mean that there is an increase in acetylation and gene activation?

l. 189 'These Findlings clearly indicate that this molecule can be used in all sorts of stem cells' is clearly an overstatement

l. 385 The sentence: 'Adult stem cells …… are epigenetically similar to their differentiated counterparts' is unclear and possibly not fully correct. Usually during cellular differentiation there is a massive reprogramming of the epigenome. The authors should explain better what they meant.

2) The clarity of the test should be improved. In several places the authors make statements which are not followed by an explanation, which makes it difficult to understand what was meant.

l. 46-48: The sentence starting with 'The ability to incorporate...' is unclear. Why does this ability make the skeletal muscle an ideal target for stem cell therapy?

l. 103 'Pax7, Myf5 and MyoD are marked by an H3K4m3' is unclear. Do the authors mean gene promoters?

l. 103-106 are not clear. H3K4m3 does not induce expression, and activation occurs by H3K27m3? This concept is not clear, why should a repressive mark lead to activation of SCs? The authors should expand and provide a clearer explanation.

l.107 the sentence 'MyoD has been associated with HMTs' is unclear. Do the authors mean that the protein MyoD interacts with several HMTs or that the gene is regulated by several HMTs?

l. 120 The sentence 'Furthermore, acetylated histones allow….' is unclear, the authors should expand and introduce the concept better.

l.170 The use of 'this' is unclear…

l. 184 It is unclear what the authors mean by 'recruitment of C2C12 myoblasts into myotubes'

l. 198 is unclear

l. 225 'can skew …. downstream' is unclear

l. 371 the connection of this chapter to the strategies for MD is unclear here, maybe the authors could cite on of two examples from the field?

3) A list of abbreviations should be included to facilitate understanding for the general reader

l. 159 What are SRF and MEF2?

l 166 What is Sgcb-null mice?

l. 251 What is SCID-X1?

l. 298 what is BMD (Becker MD?)

l. 306 What are mdx mice?

4) Legend of Figure 1 should be expanded to explain the pictures or the individual panels should be referred to in the text

5) The article suffers from multiple grammatical mistakes (see below for some examples, many more are present), the language should be carefully revised.

l. 9 'is considered as an ideal target' - 'as' should be removed

l.10 'the satellite cells, a muscle stem cell that' - should be replaced by 'the satellite cells, muscle cells that' for consistency

l. 19 'in' is missing at the end of the line

l.21 'made' is missing after been

l. 43 'the SCs usually comprises two states' What is meant by states, two sub.-populations?

l. 57 'caused by a mutation': mutations might fit better here

l.68 'despite encouraging results preclinical' should be changed to 'despite encouraging preclinical results'

l.69-70 'their' is used wrong here, the sentence should be changed to 'room to enhance the myogenic potential of MABs'

l. 71 'interested' should be changed to 'interest'

l.82 it is unclear what is meant by 'them' Do the authors mean DNA? 

l. 100 'H3K27m3/H3K27m3' should be changed to 'H3K27m3/H3K9m3'

l. 101 the activity cannot be actively expressed, this sentence should  be corrected

l. 175 'A first ...' should be replaced by 'The first'

l. 189 'these' should be changed to 'this'

l. 206 use of 'for' in unclear, do the authors mean 'thanks to'?

l.219 by is missing after iPSCs

l. 223 'have reported' should be changed to 'has reported'

l. 226 'appointed at' should be replaced by 'pointed to' and 'involved' should be removed

l.229 'to target' should be replaced by 'targeting of'

l. 236 mistake of formatting of the reference

l. 297 'and exon' should be replaced by 'an exon'

l. 307 'remained' should be changed to 'remain'

l. 316 'of' should be removed 

l. 346 'of' is missing after insertion

l. 355 'a' is missing before 'reproducible and highly efficient'

l. 388 'with' should be removed after facing

5)The introduction is quite full of specialized terminology, inclusion of a figure to illustrate main stages and players during muscle development would be helpful. Also understanding of the mutations in dystrophin gene could be facilitated with a schematic figure of the gene, most common mutations and the effect on the protein.

Author Response

Reviewer#1

The article by Breuls at al. provides a review of the epigenetic regulation of muscle stem cells and the potential use of epigenetic and genetic engineering approaches to develop stem cell therapy for muscular dystrophies. The subject of this review and the perspective given by the authors is of great interest for a broader lectureship of researches working in the field of cellular therapies, but also for a more general public.

However, the paper has several important shortcomings, including generalisations, overstatements and even scientifically incorrect statements that should be corrected. In addition, the use of specialised language and lack of explanations for several abbreviations makes the understanding of several parts of the article difficult. Finally, the language needs extensive editing, as many mistakes are present. The review would also greatly benefit from more figures to visualise the concepts and details. In addition, the authors concentrate on the role of histone modifications and miRNA, but skip DNA methylation, the dynamic role of which in both myogenesis and muscular pathologies is well established. This should be either included (at least in a few sentences) or clearly stated that is out of scope.

Detailed points:

1)    Several scientific statements are severely misleading or simply not true. The authors should  also avoid misleading generalisations.

l. 81 DNA can be methylated, but is not acetylated! Acetylation occurs on lysine residues of histones in the chromatin.

We thank the reviewer to point out this mistake. We corrected the sentence mentioned above in the revised manuscript.

l.85 Histone methylation is not by definition repressing, the activating of repressing function of the methylation of the histone tails is context dependent (e.g. H3K4me3 is activating, while H3K9me3 is repressive). Similarly, miRNA can also activate gene expression (see for example PMID: 28639193). The authors should avoid misleading generalisations. 

We agree with this reviewer that it was misleading to use methylation in general as an example of a repressive mark. Therefore, I specified this example to DNA methylations, as these are seen as repressive marks.

l.94-95 'Methylation introduces a methyl group to the cytosine' in the context of histone methylation is WRONG. Cytosines are methylated in the DNA, in histones methylation occurs on lysine and arginines.

l. 97-98 DNMTs and TET enzymes are responsible for the methylation of DNA and NOT histones. Histones are methylated by histone (or protein) lysine/arginine methyltransferases and demethylated by histone (or protein) demethylases.

Indeed, there was a mix up between the methylation that occurs on the DNA and the methylation marks that are present on the histones. The issues mentioned above were corrected and an extra chapter was added, specifically focussing on DNA methylation for sake of clarity as follows:

2.1.1 DNA methylation 

DNA Methylation introduces a methyl group to the cytosine residue, thereby causing steric hindrance, which prevents DNA-binding proteins from binding. The two groups of enzymes responsible for these methylations are the DNA methyltransferases (DNMTs) and the ten-eleven translocation methylcytosine dioxygenase (TET) family of proteins. DNA methylations usually occur in CpG islands, areas of the genome rich in cytosine residues followed by guanine residues found, often near regulatory elements [18,19]. Usually, the hypermethylation of promoter regions is associated with gene silencing while hypomethylated regions are areas of open chromatin state. A study that described the methylome changes during myobalst development, has reported the occurrence of hypermethylation waves during skeletal muscle‐lineage commitment. Those regions remained stably methylated, whereas muscle‐regulatory regions, including the master gene Myf5, underwent hypomethylation to allow the onset of the myogenic differentiation program [20]. Global DNA methylation is increased during the progression from myoblasts to mature myotubes. Furthermore, myogenic stem cell activation and differentiation is marked by the upregulation of specific DNMT isoforms [21].

l. 123 'In general, the genome is in a deacetylated state'. This is again a generalisation. Do the authors mean that there is an increase in acetylation and gene activation?

In order to avoid generalisations, we focussed specifically on MyoD and stated the following:

During SC quiescence, MyoD activity is prevented through the binding of repressor Snai1/2 to the promoters of multiple MRFs and the subsequent recruiting of HDAC 1 and 2, rendering MyoD in a hypo-acetylated state (Figure 1c) [35,36].

We hope that now it is stated correctly.

l. 189 'These Findlings clearly indicate that this molecule can be used in all sorts of stem cells' is clearly an overstatement 

The conclusion from this part has been changed as follows:

These findings indicate that these molecules can boost the myogenic potential of several sorts of stem cells in order to increase their in vivo efficacy.

l. 385 The sentence: 'Adult stem cells …… are epigenetically similar to their differentiated counterparts' is unclear and possibly not fully correct. Usually during cellular differentiation there is a massive reprogramming of the epigenome. The authors should explain better what they meant.

We rephrased the sentence as follows:

Adult stem cells, including SCs and MABs can transfer some epigenetic marks to their differentiated counterparts but are extremely rare in skeletal muscle.

2)    The clarity of the test should be improved. In several places the authors make statements which are not followed by an explanation, which makes it difficult to understand what was meant.

l. 46-48: The sentence starting with 'The ability to incorporate...' is unclear. Why does this ability make the skeletal muscle an ideal target for stem cell therapy?

To make it clear, we added some further explanation as follows:

This ability to incorporate mononucleated myoblasts into the multinucleated fibres makes the skeletal muscle an ideal target for stem cell therapy to treat muscle disorders [4-6]. In fact, when a genetically corrected stem cell fuses with the existing myotube, this feature will be transferred within the long multinucleated syncytial tissue.

Naturally, this is not the case for example in the brain, where all mononucleated cells need to be replaced for cell therapy to be effective.

l. 103 'Pax7, Myf5 and MyoD are marked by an H3K4m3' is unclear. Do the authors mean gene promoters?

We rephrased the sentence as follows:

In addition, the transcription start sites of Pax7, Myf5 and MyoD are marked by an H3K4m3.

l. 103-106 are not clear. H3K4m3 does not induce expression, and activation occurs by H3K27m3? This concept is not clear, why should a repressive mark lead to activation of SCs? The authors should expand and provide a clearer explanation.

In general, activation of satellite cells is obtained by the downregulation of Pax7. In this view, the repressive H3K27m3 mark allows the downregulation of Pax7, resulting in myogenic differentiation. 

l.107 the sentence 'MyoD has been associated with HMTs' is unclear. Do the authors mean that the protein MyoD interacts with several HMTs or that the gene is regulated by several HMTs?

Indeed, MyoD interacts with several HMTs and is also regulated itself by some of them. This is explained after this general statement as follows:

Master regulator MyoD has been associated with several HMTs, thereby regulating the expression of downstream MRFs (Figure 1). For example, in activated SCs HMT G9a is recruited to MyoD by homeoprotein Msx1 leading to the enrichment of H3K9me2 marks [31,32]. Although it represses MyoD, its expression is not abolished completely (Figure 2A). When MyoD binds to the MRF promotors, co-binding with HTMs KMT1A and Suv39h1 leads to a repression of premature activation of myogenic genes by stimulating repressive histone marks (Figure 2B) [33-35]. Upon differentiation, MyoD can be recruited to the promoters of several MRFs through the increased accessibility caused by the SWItch/Sucrose NonFermentable (SWI/SNF) chromatin-remodelling complex [36,37]. Once differentiation is induced, MyoD together with HMTs Set7/9 and DN-JMJD2A leads to the removal of repressive histone marks and the acquisition of H3K4m3 (Figure 2D) [38,39]

l. 120 The sentence 'Furthermore, acetylated histones allow….' is unclear, the authors should expand and introduce the concept better.

The explanation of histone acetylation was adapted as follows:

Acetylation of the lysine residues on the histones neutralizes the positive charge, thereby relaxing the chromatin. By relaxing the chromatin, transcriptional binding sites become more accessible, leading to increased gene expression.

l.170 The use of 'this' is unclear…

We replaced ‘this’ in the sentence as follows:

Due to the increasing knowledge in epigenetic regulation of myogenesis, researchers started to consider the adaptation of epigenetic marks as a strategy to reprogram the stem cell fate.

l. 184 It is unclear what the authors mean by 'recruitment of C2C12 myoblasts into myotubes'

By the exposure to an HDACi, C2C12 myoblast form better myotubes through the fusion of these mononucleated cells. To make it clearer in the text, we changed the sentence as follows:

More specifically, in vitro, it has been shown that these different HDACi increase the fusion of C2C12 myoblasts into myotubes [58].

l. 198 is unclear

The sentence was changes as follows:

Furthermore,in C2C12 cells,the treatment with 5AC resulted in enhanced expression of muscle-specific genes (including myogenin), increased myotube maturation and spontaneous contraction, suggesting that inhibition of methylation could be a suitable tool to further boost myogenic differentiation in already committed precursors.

l. 225 'can skew …. downstream' is unclear 

In the hope to make it clearer, the sentence was modified as follows:

A study conducted on iPSC differentiation has reported that members of the miR-290 cluster (miR-291-3p, miR-294, and miR-295) are directly involved in murine iPSC reprogramming, and additionally can influence cardiomyogenic lineages maturation [83].

l. 371 the connection of this chapter to the strategies for MD is unclear here, maybe the authors could cite on of two examples from the field?

Although the included references pointed to examples in the field of MD, this chapter was written in a very general way. To improve the connection with the rest of the text, we adapted the paragraph by explaining the examples in more detail as follows:

The integration of suicide genes is of great importance when using PSCs, as differentiations are often impure leading to risk for teratoma formation [139]. Suicide genes are used as safety switches in order to intervene when pluripotent stem cells derivatives undergo excessive proliferation. So far, this system has been mostly investigated on a PSC-level, making its use to specifically eliminate cells from the skeletal and cardiac muscle less explored. The first suicide gene used in the area of MD, is the inducible Caspase 9 system, an enzyme that induces apoptosis upon administration of a chemical inducer of dimerization. This system was used 8 weeks after injection of mesodermal progenitors into the skeletal muscle, showing complete clearance of the injected stem cells [140]. Within the cardiac muscle, the incorporated human somatostatin receptor 2 was incorporated, which made it possible to eliminate the engrafted cells upon the administration of 177Lu-DOTATATE, a β--emitting radioisotope specifically binding the integrated receptor. The benefit of this gene is that, just by adding a different radionuclide, it can also serve as a reporter gene. This means that by integrating a single gene, the injected stem cells can be monitored and, upon observation of an abnormal growth, can be eliminated [141].

3) A list of abbreviations should be included to facilitate understanding for the general reader

All the abbreviations mentioned by the reviewer were clarified further in order to make the text more understandable. Furthermore, a list of abbreviations was included in the revised manuscript.

4) Legend of Figure 1 should be expanded to explain the pictures or the individual panels should be referred to in the text.

All subpanels are now referred to in the text. Furthermore, we adapted the legend to make the figure easier to interpret. The legend Figure 2 in revised manuscript now states the following:

Figure 2.Epigenetic regulation of MyoD during satellite cell quiescence and activation. MyoD is repressed during satellite cell quiescence due to the presence of H3K9me2 marks, recruited by the Msx1/G9a complex (A). Furthermore, the binding of MyoD to the muscle regulatory factors (MRFs) through the binding of Snai1/2:HDAC1/2 complex, rendering the MRFs hypo-acetylated. The small amount of MyoD that binds to the MRFs will recruit SUV39h and KMT1A leading to the addition of repressive H3K9/27 marks (B). Upon satellite cell activation, repressive marks are removed from both MyoD, by Kdm1a and the MRFs by phosphor-retinoblastoma (pRb). Furthermore, acetylation is increased by binding of p300 and pCAF (C and D). Once MyoD is activated it will bind to the MRFs and recruit Set7/9, JMJD2A and SWI/SNF that will add permissive H3K4 methylation marks (D)

5) The article suffers from multiple grammatical mistakes (see below for some examples, many more are present), the language should be carefully revised.

We corrected grammatical mistakes throughout the manuscript and, when necessary, we modified the according to the reviewer’s suggestions. 

6) The introduction is quite full of specialized terminology, inclusion of a figure to illustrate main stages and players during muscle development would be helpful. Also understanding of the mutations in dystrophin gene could be facilitated with a schematic figure of the gene, most common mutations and the effect on the protein.

To make the text more understandable, two figures were added to the paper: one concerning the myogenesis (Figure 1 in revised manuscript) and the other one pointing out the most common mutations in the dystrophin gene (Figure 3 in revised manuscript).

Reviewer 2 Report

        The authors reviewed progress in genetic/epigenetic alteration in myogenic stem cell and their potentency usage in clinical. This review is well organized and gave an adequate references to related work. But I still have some concerns about the present version.

 1. There are many minor mistakes in this review. Such as in line 100, H3K27me3/H3K27me3 should be corrected as H3K27me3/H3K9me3. In line 142,  the cleaved shorter precursor should be called pri-miRs, not pre-miRs. In line 203, it should be reordered to 2.2.2 miRNA modulations. In 236, reference from Quattrocelli et al should be in correct format. The sentence in lin 319-320 is not a whole sentence. line 381. it should be re ordered to 4. Conclusions and future directions.

2. The abbreviations for the techniques in Table 1 should be listed below the table.

3. Line 370, more detail should be given to this study(ref.134)and the authors should discuss its influence.

4. The format of this review should be checked carefully. Such as some paragraphs have indent, but other didn't. 

Author Response

The authors reviewed progress in genetic/epigenetic alteration in myogenic stem cell and their potentency usage in clinical. This review is well organized and gave an adequate references to related work. But I still have some concerns about the present version.

 1. There are many minor mistakes in this review. Such as in line 100, H3K27me3/H3K27me3 should be corrected as H3K27me3/H3K9me3. In line 142,  the cleaved shorter precursor should be called pri-miRs, not pre-miRs. In line 203, it should be reordered to 2.2.2 miRNA modulations. In 236, reference from Quattrocelli et al should be in correct format. The sentence in lin 319-320 is not a whole sentence. line 381. it should be re ordered to 4. Conclusions and future directions.  

Dear reviewer, thank you for indicating these mistakes. All of them were adapted.

2. The abbreviations for the techniques in Table 1 should be listed below the table.

The following abbreviations were added to the table:

ZFN: zinc-finger nucleases; TALEN: transcription activator-like effector nucleases; CRISPR: Clustered Regularly Interspaced Short Palindromic Repeats; Cas: CRISPR associated proteins; Cpf: CRISPR from Prevotella and Francisella 1; Δ: deletion; Ex: exon; Dupl: Duplication; Pt: point mutation

3. Line 370, more detail should be given to this study(ref.134)and the authors should discuss its influence.

We elaborated on this study as the reviewer requested. The text was modified as follows:

In order to move this application towards the clinic, site-specific integration is essential. So far However; in this field, only one report shows the integration of these reporter genes in a site-specific manner [138]. More specifically, Holvoet et al. integrated the human sodium iodide symporter into the AAVS1 locus of embryonic stem cells through the use of ZFN. This integration led to a stable expression of the reporter over time without influencing the functionality and differentiation potential of the cells. By combining targeted genome-engineering with stem cell imaging, the transition of stem cell therapy to a clinical setting will be greatly accelerated.

4. The format of this review should be checked carefully. Such as some paragraphs have indent, but other didn't.

The format was made consistent over the entire review. 

Round  2

Reviewer 1 Report

The current version of the manuscript is considerably improved. The authors took care to explain the concepts more clearly and corrected the generalisations and misleading statements carefully. The addition of the two figures greatly improves conceptualization. I only have one final comment regarding the new paragraph about DNA methylation (staring in line 102)

The statement that CpG methylation usually occurs in CpG islands is misleading, as CpG islands of many genes are unmethylated. It is true that they may become methylated, for example in cancer and this will usually lead to the silencing of the gene. In addition, not all promoters have CpG islands. I would suggest to remove that sentence completely or explain that CpG islands are considered regulatory regions for DNA methylation, as their methylation often leads to gene silencing.

Author Response

We thank Reviewer#1 for all comments and to recognise our efforts to improve the ms according his/her suggestions.  Finally, we adapted the sentence regarding CpG island as follows:

CpG islands, areas of the genome rich in cytosine residues followed by guanine residues, are considered regulatory regions for DNA methylation. Although not all promoters have CpG islands, the hypermethylation of these regions is associated with gene silencing while hypomethylated regions are areas of open chromatin state. 

Best wishes,

Maurilio, Giorgia and Natacha

Reviewer 2 Report

The review is much better than its previous version. And I think it can be accepted now. 

Author Response

We really thank Reviewer #2 for his/her comments and suggestions that certainly contributed to improve our ms.

Cells EISSN 2073-4409 Published by MDPI AG, Basel, Switzerland RSS E-Mail Table of Contents Alert
Back to Top